# Understanding the roles and work of paramedics in primary care: a national cross-sectional survey

Georgette Eaton ![ORCID],[1] Stephanie Tierney,[1] Geoff Wong,[1] Jason Oke,[1] Veronika Williams,[2] Kamal Ram Mahtani[1]

¹Nuffield Department of Primary Care Health Sciences, Oxford University, Oxford, UK
²Faculty of Education and Professional Studies - School of Nursing, Nipissing University, North Bay, Ontario, Canada

**Correspondence to**
Georgette Eaton;
georgette.eaton@phc.ox.ac.uk

## ABSTRACT

**Objectives** This research aimed to fill a current knowledge gap, namely the current scope of clinical role of paramedics in primary care, in relation to specific constructs such a level of education and clinical experience.

**Setting** The survey was distributed to paramedics in primary care across the UK through the College of Paramedics.

**Participants** A total of 341 surveys were returned (male=215). 90% of responses were from paramedics in England, 1.7% from paramedics in Northern Ireland, 4.6% from paramedics in Scotland and 2.9% from paramedics in Wales. This represents approximately 33% of the primary care paramedic workforce in England and Wales. Estimates for percentages in Northern Ireland and Scotland are unavailable due to the lack of workforce datasets capturing paramedics in primary care.

**Results** Considerable variation was found in job titles, level of education and provision of clinical supervision of paramedics in primary care. Differing levels of practice were noted, despite guidance documents that attempt to standardise the role. Statistical analysis of quantitative data highlighted that relationships exist between paramedic clinical exposure in primary care, level of education, and ability of independently prescribe medicines and the extent to which clinical presentations are seen and examinations performed. However, free-text responses indicated that challenges in relation to access to further education and clinical supervision to support clinical development resulted in frustration for paramedics who work in this setting.

**Conclusions** As well as offering an insight into the demographics of the primary care paramedic work force, there is indication of the clinical scope of role undertaken in this setting. Based on our findings, we recommend changes to education and support, governance and legislation to ensure paramedics employed in primary care can work to achieve the full extent of their professional capability.

## STRENGTHS AND LIMITATIONS OF THIS STUDY

⇒ The survey was distributed by the UK College of Paramedics and on social media, and so may not capture the experiences of paramedics not members of their professional body or without profiles on social media.
⇒ There was a low uptake in responses from paramedics working in Northern Ireland and Scotland.
⇒ All (n=341) respondents answered all questions within the survey.
⇒ The sample size power calculation was met, assuring an adequate power to detect statistical significance in the results.
⇒ The survey was undertaken during September–November 2021, during which time UK primary care was continuing to respond to the ongoing global pandemic of COVID-19.

## INTRODUCTION

Over the last decade, paramedics in the United Kingdom (UK) have increasingly taken up clinical employment away from ambulance services, with many moving into primary care settings.[1] Reasons for this move are multifactorial and interwoven. Changes to healthcare access have created a sociocultural dependence on the ambulance service,[2] resulting in an increase of urgent and primary care related dispositions attended by paramedics.[3] To respond to the changing demands of 999 calls, paramedic preregistration education prepares registrants to deliver holistic care across the lifespan,[4] moving beyond protocolised training for the management of emergency presentations, for which paramedics have historically been most well known. This broad undergraduate education defines the pluripotential nature of paramedics, enabling them to act as generalist clinicians in a range of clinical settings—and ultimately enables them to leave the ambulance service. This, coupled with poor managerial support and a lack of clinical progression within the ambulance service,[5] is resulting in paramedics seeking opportunities to work in other clinical settings which offer further development and an improved work–life balance.[6] Occurring at a time where the primary care workforce is understaffed due to a failure to recruit and retain general

practitioners (GPs),[7] paramedics actively seek opportunities to work in primary care, believing their capabilities fit well within this clinical setting.[8] Indeed, recent National Health Service (NHS) workforce policy welcomes this professional group into primary care,[9–11] with associated funding through the Additional Roles Reimbursement Scheme available for practices employing paramedics (among other health and social care staff) in England.[12]

Our previous research has highlighted that for paramedics to work successfully as part of the primary care team, transition into this workforce needs to be supported through a combination of formal education, clinical supervision and socialisation.[8] Desired outcomes, such as increasing clinical capacity within the primary care team (and filling workforce gaps created by the shortage of GPs), may then transpire. Health Education England (HEE) has made steps to provide a framework that addresses these concepts. HEE is a non-departmental public body, which provides coordination and support for the training and education within England's healthcare workforce. In 2019, HEE set out a 'roadmap' for paramedics to follow as they transition into primary care roles.[13] This document outlines the qualifications, capabilities, clinical skills and case presentations paramedics are expected to encounter while working in primary care. This is the only published document across the devolved nations that sets any scope of role for paramedics working in this clinical setting.

Paramedics are being actively recruited into the primary care workforce, but knowledge gaps remain. This research specifically aims to fill a current knowledge gap, namely the paramedics' scope of role in primary care and their perception of their roles. This research records the current scope of clinical role of paramedics in primary care, in relation to specific objectives:

► To better understand the patterns of education level, experience, salary, prescribing status and clinical supervision for paramedics in primary care.
► To investigate the scope of role undertaken by paramedics in NHS primary care.
► To explore the perceptions paramedics in primary care have on their contribution to primary care teams.

## METHODS
Prior to commencing this survey, a study protocol was developed and registered with OSF Registries (10.17605/OSF.IO/YKDA7). We report our findings according to the Strengthening the Reporting of Observational Studies in Epidemiology statement.[14]

### Study design
An online survey was distributed via the College of Paramedics to paramedics in primary care in England, Northern Ireland, Scotland and Wales. The survey used both qualitative and quantitative items in recognition of the polygonal objectives of the study.[15] Details regarding workforce data for paramedics in primary care are reported for England[16] and Wales,[17] but not yet for Northern Ireland or Scotland. Based on the available data in England and Wales, we estimated that approximately 1500 paramedics were working in primary care roles across the UK in August 2021, and therefore, considered a sample size of 306 would be needed for a CI of 95% and margin of error of 5% for statistical analysis.

### Materials and procedure
The survey was developed by authors in consultation with members of a patient and public involvement group, and stakeholders from the College of Paramedics, HEE and the Royal College of General Practitioners. The survey was first piloted with paramedics within this stakeholder group and refined by the authorship team. This pilot assisted in the flow of questions, and the format of question presentation. The final survey was presented using Jisc online surveys (https://www.onlinesurveys.ac.uk/) (online supplemental file 1). Data collection took place between 1 September 2021 and 30 November 2021.

Distribution of the survey was initially via internal communications within the College of Paramedics, as well as across their social media platforms. This survey was shared further on social media to reach paramedics working in primary care who may not be members of the College of Paramedics.

Respondents were offered a £10 Amazon e-voucher in compensation for their time to complete the survey.

### Data analysis
Free-text responses were analysed using semantic level, inductive thematic analysis in NVivo V.12,[18] undertaken by GE. Ten per cent of these codes were reviewed by ST. Quantitative data were analysed using descriptive statistics (mean, SD and frequencies) and appropriate non-parametric tests ($\chi^2$ test of independence, Kruskal-Wallis Test, Mann-Whitney test, Spearman's r correlation) in IBM SPSS Statistics, V.28 (IBM). Bonferroni correction was used to counteract problems where multiple comparisons occurred, and this adjustment is included in the results reported in this paper. Statistical analysis was undertaken by GE and reviewed independently by JO.

Quantitative and qualitative data were initially analysed separately, then merged and interpreted.[19] During interpretation, data were considered in relation to existing conceptual frameworks[8] to enrich findings and synthesise complementary results.

### Patient involvement
The patient and public group associated with this research were involved during the design of the survey instrument and the analysis of the qualitative results.

## RESULTS
A total of 341 responses were returned, of which 90.6% were from paramedics working in primary care in England (n=309). Based on workforce data published for the period within which the survey was undertaken

**Table 1** Description of survey respondents

| Age range | n | Gender | n | Country of work | n |
|---|---|---|---|---|---|
| 18–24 years old | 4 | Female | 126 | England | 309 |
| 25–34 years old | 98 | Male | 215 | Northern Ireland | 6 |
| 35–44 years old | 113 | | | Scotland | 16 |
| 45–54 years old | 96 | | | Wales | 10 |
| 55–65 years old | 30 | | | | |

(September–November 2021),[16][17] our respondents represent 33% of the population for England and Wales separately. Table 1 outlines descriptive characteristics of survey respondents according to age range, gender and country of work. All respondents answered each question in the survey.

### Experience as a paramedic prior to working in primary care

Prior to working in primary care, most respondents had 3–5 years (n=93) or 6–10 years (n=93) experience of working as a paramedic. When asked why they chose to work in primary care, a majority cited an improved work–life balance as the single biggest contributing factor. This was often referred to in relation to a move away from the unsustainable workload in the ambulance service, especially working hours that were considered family friendly, with no night or weekend shifts. Other factors prompting paramedics to take up employment in primary care included the opportunity for clinical development; increased job satisfaction associated with the increased autonomy and enhanced clinical skill set required to work in primary care; and a clinical interest in holistic and preventative medicine associated with general practice.

### Level of education

The majority (52%) of respondents were educated to at least one module at Framework for Higher Education Qualification (FHEQ) level 7/Scottish Credit and Qualifications Framework (SCQF) level 11, with more than half of these educated to postgraduate certificate (n=90). Figure 1 outlines the level of education for respondents. The need for additional knowledge to work in primary care was articulated. Respondents often reported a 'constant pressure to study' (RID 118; Emergency Care Practitioner), and a sense of overwhelm regarding the volume of academic work alongside their clinical workload. Both a lack of protected time to study, as well as absence of funding from primary care employers for paramedics to undertake education qualifications, were highlighted as a hinderance to development in primary care.

### Effect of HEE roadmap

For respondents in England, the 'roadmap' published by HEE[13] was viewed to assist in the standardisation and supervision of the role (n=42), but was often poorly executed at an employment level, due to lack of clarity in the roadmap's requirements (n=21), ease of fulfilment (n=25) or uptake by employer (n=21). Fifty-seven respondents reported the roadmap had no influence on their employment or practice in primary care:

> I'm not ARRS funded so doesn't impact me. I'm aware of the roadmap but my surgery isn't pressuring me to complete it. They know my skill set. (RID 327; Paramedic Practitioner).

### Clinical supervision

85.6% (n=292) of respondents received clinical supervision within their role. When asked to what extent the clinical supervision meets their needs, 25 respondents reported a clinical supervision model that was regular and structured. Other experiences of supervision were 'very poor due to limited time with supervisor' (RID 51; Paramedic Practitioner).

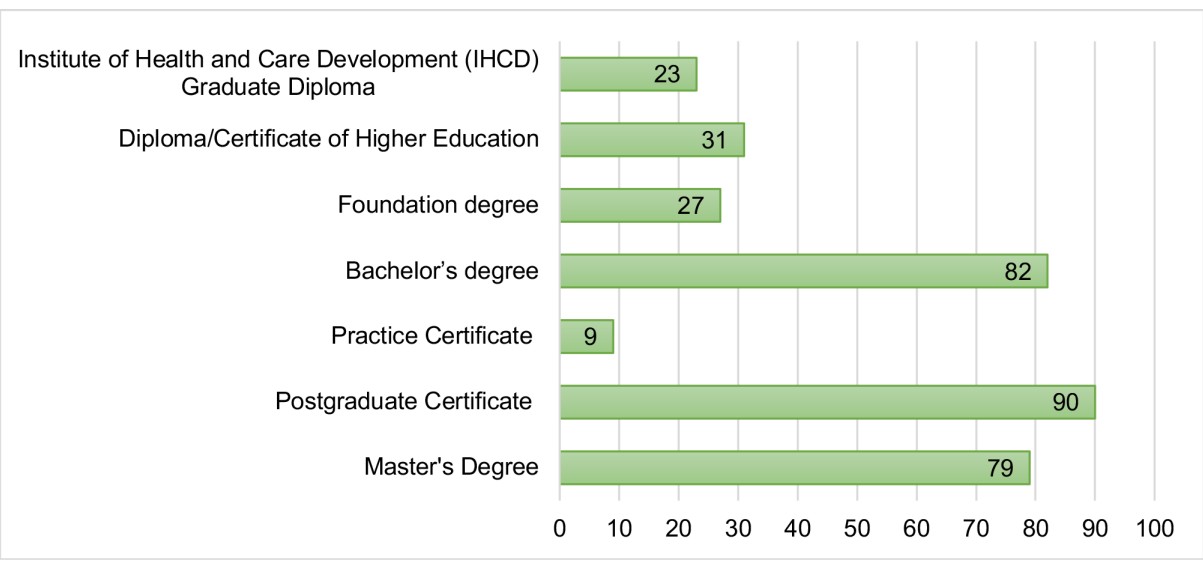

**Figure 1** Level of education of respondents.

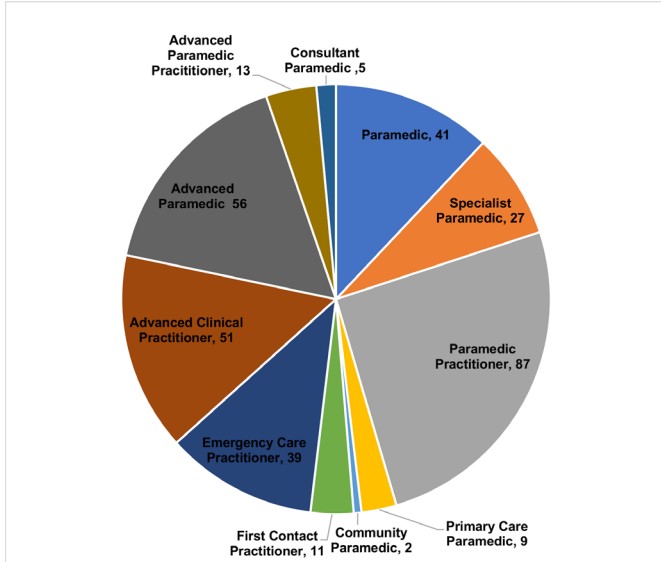

**Figure 2** Job title of respondents.

I don't feel the GPs really understand the role yet or have the time to support it in a way that would meet the expectations set out by the… roadmap. (RID 199; Paramedic Practitioner).

While a trend was seen concerning the provision of clinical supervision and the extent to which clinical presentations were seen and clinical examinations undertaken, this failed to reach statistical significance (online supplemental tables A and B).

### Job title

Job titles for paramedics working in primary care were varied and inconsistent (figure 2). Inconsistency in job titles was suggested to contribute to discrepancies in scope of practice and understanding of the role from other healthcare professionals in primary care:

I still feel that there is a lack of understanding, or clear delineation between roles… I am unsure if those at my surgery are aware of the difference between a Paramedic Practitioner and Advanced Clinical Practitioner… (RID 305; Advanced Clinical Practitioner).

As well as patients:

I don't think [my job title] reflects my role and is unclear to patients. (RID 150; Emergency Care Practitioner).

It appears that job title (and thus seniority of paramedic) made a difference to the extent to which a diagnosis is made during a consultation ($H^3$=15.73, p≤0.001), the management of medical and clinical complexity ($H^3$=15.73, p≤0.001), and leadership and management ($H^3$=10.507 p=0.015) undertaken by paramedics.

### Salary

The most common salary bracket was between £33 222 nd £43 041 (figure 3). Significant associations were found between salary and job title ($\chi 2$ (15)≥51.137, p≤0.001), prescribing status ($\chi^2$ (5)≥118.190, p≤0.001), highest qualification ($\chi^2$ (30)≥72.589, p≤0.001), length of time as a paramedic ($\chi^2$ (40)≥101.212, p≤0.001) and length of time in primary care ($\chi^2$ (25)≥112.780, p≤0.001).

There was no significant association between salary and gender ($\chi^2$ (5)=11.936, p=0.036) or salary and region of work within the UK ($\chi^2$ (15)=28.589, p=0.018).

### Length of time working in primary care

Length of time in primary care differed with job title ($H^3$ =14.145, p=003), where seniority of job title was associated with increased time in primary care. Elements of the clinical work associated with length of time in primary care are reported in online supplemental tables A and B.

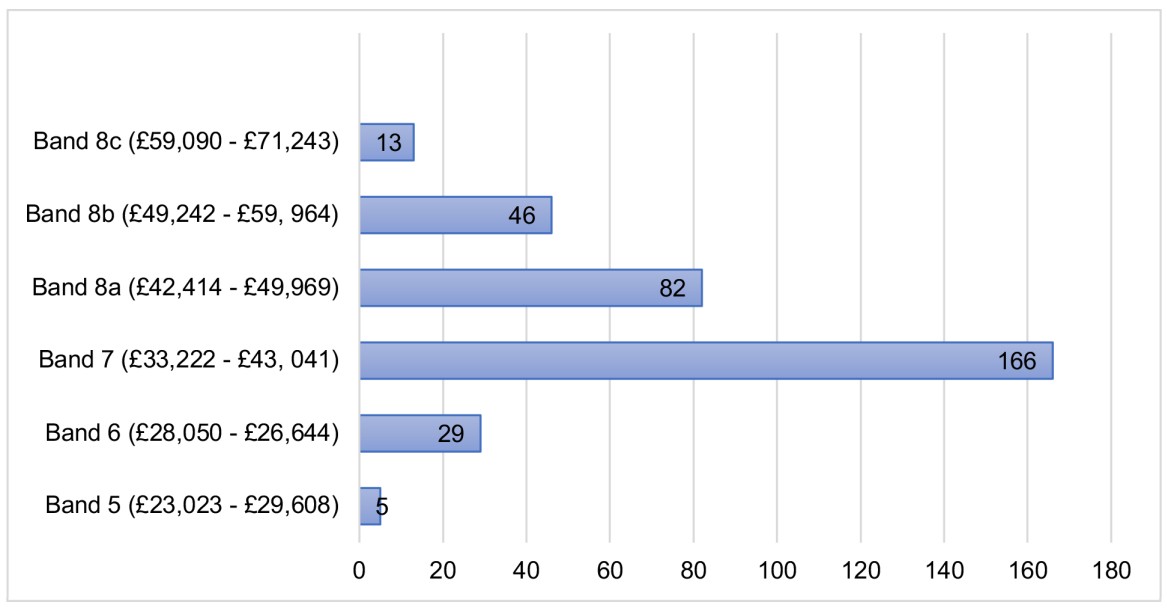

**Figure 3** Salary of respondents.

### Hours worked in primary care

Significant associations were found between hours worked and employment type ($\chi^2$ (15)≥129.872, p≤0.001), where there was an association found between respondents employed directly with the practice and working full time hours (n=126), and those working in rotational roles between ambulance services and primary care providers working 1 day a week (n=22).

### Core capabilities of paramedics working in primary care

We assessed the relationships between the extent to which the core capabilities of primary care set out within HEE's Roadmap[13] formed part of the respondent's role, against employment type, length of time registered as a paramedic, length of time in primary care, level of education, hours worked, job title, independent prescribing status and salary. Positive relationships exist between these capabilities and increase in each of these factors. The strength of these correlations are outlined in online supplemental table C.

However, some respondents experienced frustration regarding their progression within these core capabilities:

> Keen to undertake a leadership role but don't see this being an option. Very much bums on seats. (RID 92; Advanced Clinical Practitioner).

### Clinical work undertaken

To understand the breadth and frequency of clinical work undertaken, respondents were asked the extent to which they saw the range of clinical presentations (figure 4) and the extent to which they undertook clinical examinations (figure 5) outlined in HEE's Roadmap.[13]

Relationships between clinical presentations existed between length of time as a paramedic, length of time in primary care, hours worked in primary care, highest qualification, job title and prescribing status. Relationships between clinical examinations undertaken also existed between length of time in primary care, hours worked in primary care, highest qualification, job title and prescribing status. These results are outlined in online supplemental tables A and B.

### Blood tests

The request and interpretation of blood tests by paramedics in primary care was positively influenced by prescribing status, the receipt of clinical supervision, increased salary, hours worked (full time) and higher levels of education—however, these results largely failed to meet statistical significance. There was no correlation between the request and interpretation of blood tests and length of time as a paramedic, nor length of time in

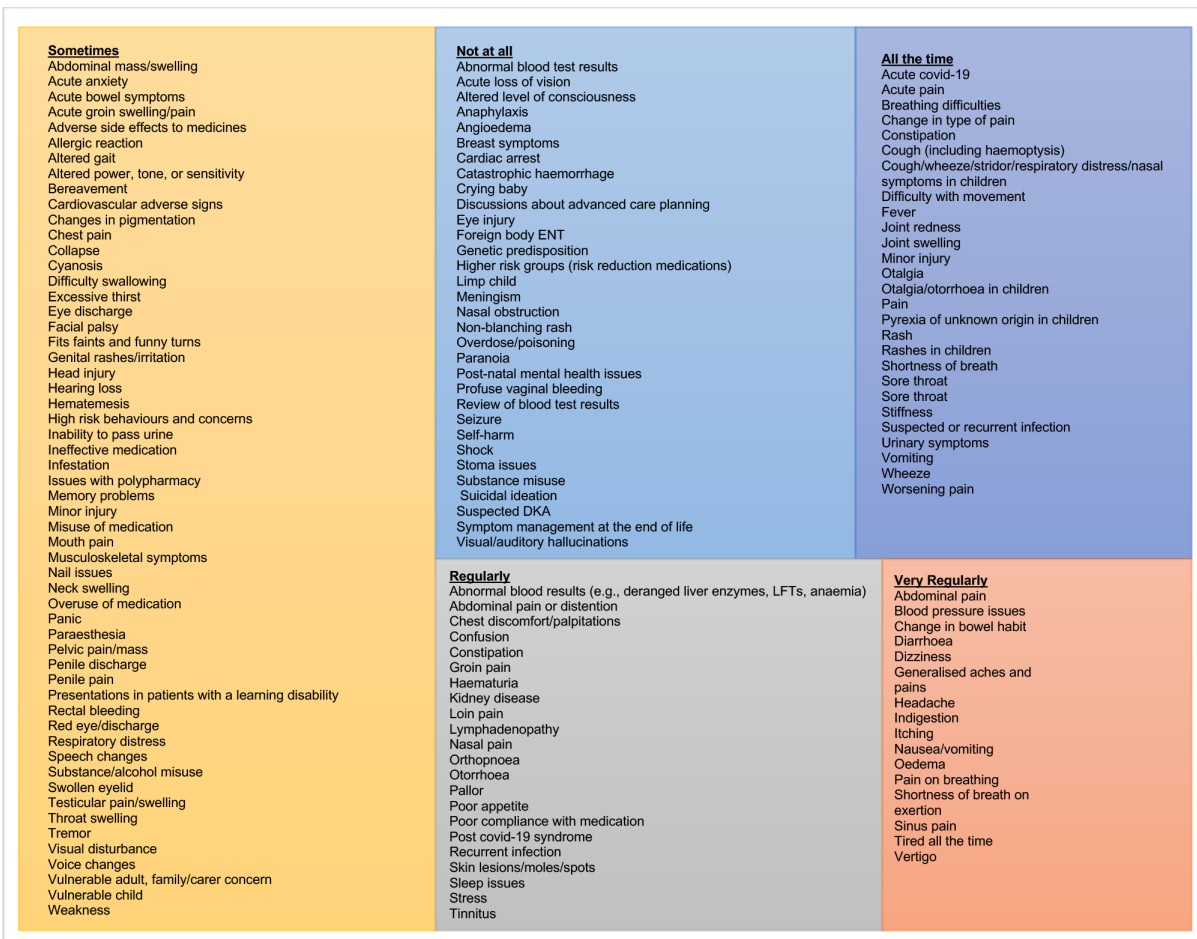

**Figure 4** A treemap of clinical presentations attended by respondents as a paramedic in primary care.

**Figure 5** A treemap of clinical examination undertaken by respondents as a paramedic in primary care.

primary care. These results are outlined in online supplemental table D.

### Patient groups not seen

When asked about patient groups not seen, many respondents (n=189) would not see presentations relating to women's health (including intimate examination, pregnant patients with directly related pregnancy issues, sexual health or menopause). Other common patient groups not seen by paramedics included children, particularly under the age of 2 (n=114), and presentations relating to mental health (n=69). In addition, respondents saw clinically complex patients, palliative care and chronic conditions less than other patient groups.

### Emergency skills

Significant correlations were found between level of education and the undertaking of emergency procedures ($r_s$=−0.156, p≤0.004), where respondents with lower qualifications undertook emergency procedures to a greater extent. A similar correlation was found between the level of education and attending presentations such as cardiac arrest ($r_s$=−0.106, p=0.049), catastrophic haemorrhage ($r_s$=−0.119, p=0.028), shock ($r_s$=-0.173, p=0.001), cardiovascular adverse signs ($r_s$=−0.120, p≤0.027) and limp children ($r_s$=0.125, p=0.021), again where respondents with lower qualifications attended these presentations to a greater extent. This was also reflected in the free-text comments, where some paramedics '…*do not feel challenged*

*by only dealing with same day emergency consultations.*' (RID 14; Paramedic).

There was also a correlation between hours worked and attending presentations such as catastrophic haemorrhage ($r_s$=0.109, p=0.044), anaphylaxis ($r_s$=0.127, p=0.019), angioedema ($r_s$=0.140, p=0.009), seizures ($r_s$=0.147, p=0.007) and overdose/poisoning ($r_s$=0.200, p≤0.001), where respondents who worked less hours in primary care (such as 1 day a week or 10–20 hours per week) attended these presentations to a greater extent. It was also noted that these respondents were employed primarily by ambulance services, and so likely to be working in rotational roles.

Free-text comments outlined that respondents who had been employed for a longer time in primary care missed the opportunity to attend higher acuity patients, and also reported a loss of paramedic identity without the opportunity to practice their emergency skills.

### Prescribing status

A total of 125 respondents (36.7%) were independent prescribers, with a further 57 (16.7%) undertaking the course at the time of the survey, and a further 137 (38.7%) wishing to undertake the course in the future. Inability to prescribe frustrated 20 respondents when asked about the frustrations in their role.

Significant associations were found between prescribing status and clinical assessment skills outlined in online

supplemental table E, where independent prescribers were more likely to undertake these examinations when compared with paramedics who could not prescribe.

## Role frustrations

Short appointment times and volume of patients within the day contributed to high workload and feelings of frustrations about their role in primary care. General system frustrations were also reported regarding the structure, organisation and funding of primary care. In addition, the impact of the ongoing global pandemic of COVID-19 on the working practices of respondents caused frustration, with experiences of an increase in telephone consultation and aggression from patients.

Respondents reported frustration when faced with restrictions in practice, commonly due to an inability to prescribe schedule 2 controlled medicines, or to provide patients with a Statement of Fitness for Work. Other restrictions in practice included an ability to fit contraceptive devices, as well as certain procedural skills such as undertaking intimate examination.

Another frustration reported was a lack of understanding of the capabilities of the paramedics in primary care, from both colleagues and patients:

> Sometimes there are major misconceptions about what paramedics can and can't do. Most management have no idea about the role of a paramedic. (RID 326; First Contact Practitioner Paramedic).

> I am well qualified for my role, I am experienced. But I work alongside doctors who do not understand my capabilities. (RID 15; Advanced Clinical Practitioner, Paramedic).

## Role gratification

When asked about the difference respondents felt they made in the role, this was considered between the difference they felt they made in the primary care workforce, the difference to patients and the difference their role had on the paramedic profession overall.

### Primary care

Just under half of respondents felt their role in primary care increased workforce capacity (n=161, 47%). This was in the context of freeing up time '*for GPs to manage complex conditions*' (RID 9; Advanced Paramedic Practitioner); facilitating multidisciplinary team working and being able to capitalise on their unique paramedic skills:

> Our clinical knowledge with ability to make autonomous safe decisions with the ability to recognise big sick / little sick quickly means that we are a strong spoke in the wheel of general practice. (RID 303; Specialist Paramedic).

### Patients' satisfaction

Respondents measured the satisfaction of patients who they had seen in their role by virtue of positive feedback received, and patients who actively sought consultations

with them again in the future, or did not reattend for the same presenting complaint. This patient satisfaction was attributed to the comprehensive consultations undertaken by paramedics, which they reported as benefiting from increased time slots, and the discussion of medical conditions in a simple language patients could understand. Indeed, paramedics viewed their interpersonal skills as one of their greatest strengths regarding patient satisfaction:

> For the most part people are happy to see a professional that listens to them, treats them with respect and dignity and is competent. I think a lot of paramedics offer this in primary care. (RID 135; Advanced Clinical Practitioner).

Experiences of patient dissatisfaction following consultation by a paramedic in primary care were also reported by respondents. These occurred when patients were disappointed that the consultation was not with a GP, and where there was a lack of understanding of the paramedic role, leading them to question the clinical expertise paramedics could offer:

> Some patients do not understand my role/skill set and just want a GP. (RID 143; Advanced Paramedic).

### Profession

Respondents considered their role in primary care in relation to promotion and understanding of the profession, which '*shows that there is so much more than being a paramedic on the back of an ambulance*' (RID 144; Paramedic Practitioner). Other respondents considered this in relation to the development of the profession as a whole:

> I feel as though the level of autonomous clinical decision making, management of complex cases, referrals and medicine management we are practising is at a much higher level than a large amount of the profession. I think it sets a good example to other paramedics and other medical colleagues alike that paramedics can be pushed beyond traditional roles and expectations and can prove to be valuable part of clinical team. (RID 202; Advanced Paramedic).

The impact of the move to work in primary care was also acknowledged in context of the wider healthcare workforce. Respondents outlined this both in terms of leaving the ambulance service to undertake roles in primary care, as well as rotational working between ambulance services and primary care settings. This was considered to have benefit regarding the transferability of clinical knowledge and skills from one clinical setting to the other:

> My rotational role increases understanding between ambulance & primary care, increases my knowledge & understanding which I can pass on to colleagues and increases visibility, trust and understanding of the profession throughout primary care and the lay community. (RID 36; Specialist Paramedic).

## DISCUSSION
### Main findings of this study

This research confirms previous publications which noted variance in (A) job title reported by paramedics working in primary care; (B) the clinical work and examinations undertaken by paramedics in this setting and (C) entry requirements in terms of clinical experience and education to work in primary care.[1 8] This level of variation subsequently leads to confusion around the scope and expectations for the role and contributes to a lack of recognition of paramedics within primary care teams. While attempts have been made through HEE's Roadmap[13] to outline a framework to address this, this is applicable only in England, and such a framework has no influence for paramedics across the devolved nations. The main barriers to engagement with this Roadmap were competing workload pressures affecting the delivery of clinical supervision and uptake of this framework by primary care employers. Such inconsistency contributed to frustration and demotivation among respondents, who were concerned regarding their clinical development in this setting.

### Clinical examinations and procedural skills

Our analysis indicates that length of time in primary care, higher levels of education and status as an independent prescriber all contribute to an increase in the scope of role for paramedics in primary care. Indeed, an inverse association was also observed, where paramedics with lower educational qualifications attended emergency presentations within primary care to a greater extent than those who had undertaken higher education. This outlines that, while the paramedic may transition into primary care due to the virtue of their generalist background, their productivity in primary care may be influenced by further education and feedback regarding their clinical experience in this setting. Despite this generalist background, this survey also outlines there are patient groups commonly not seen by paramedics. This could be due to a creep into the paramedic role of nursing policy which emphasise that nurses should refer women who are pregnant to midwifery or physician care if they are not dual registered in this area[20]; and legacy of instructions for paramedics in ambulance services to convey all children under the age of 2 to emergency departments, and children under the age of 5 must be seen by a physician if non-conveyed.[21]

The survey also highlighted that paramedics working less hours in primary care (such as 1 day a week or 10–20 hours per week) attended emergency presentations in their primary care role to a greater extent when compared with their full-time counterparts. Such hours are common in rotational models, where paramedics, specialist paramedics or advanced paramedics work in a split clinical role between ambulance services and primary care settings in an attempt to increase workforce capacity in primary care and reduce attrition from the ambulance service.[22] While the ability for paramedics to attend emergency presentations in primary care may be a benefit for primary care providers, this does little to develop their primary care clinical acumen.

### Paramedic taxonomy

Some job titles reported by respondents match those endorsed by the College of Paramedics[23] (such as 'Paramedic', 'Specialist Paramedic', 'Advanced Paramedic' or 'Consultant Paramedic') or those outlined by HEE[13] (such as 'First Contact Practitioner' and 'Advanced Practitioner'), yet there remains a variety of job titles that do not correlate to these archetypes.

Our analysis indicates that, as paramedics take up more senior roles in primary care (such as 'advanced paramedic'), their scope of role increases in relation to clinical examinations performed and the clinical presentations they attend. Such an increase in scope could be due to their ability to independently prescribe and undertaking postgraduate study. Independent prescribing is typically undertaken by 'advanced paramedics' who have completed (or working towards completion of) a master's degree.[24] We noted that these paramedics are more likely to make a diagnosis during the consultation and manage medical and clinical complexity. This is in contrast to 'paramedics' or 'first contact practitioners', who may have a similar scope of clinical examination, but a reduced scope in relation to managing clinical complexity and making a diagnosis. This supports previous findings where such paramedics are employed in an 'eye and ears' approach only.[8]

There was a strong correlation with advanced and consultant level roles and undertaking activities related to leadership and management in primary care. This suggests that paramedics may move into leadership roles within primary care that have traditionally been filled by GPs. However, there was no correlation regarding undertaking research activities and job title. This indicates that research activities are less accessible to paramedics in primary care, despite being a pillar of advanced practice, matching previous research findings.[25]

### Strengths and weaknesses of the study

This is the first national survey of the paramedic role in primary care within the UK. It has international relevance for primary care workforce transformation in countries where paramedics operate in a similar way to in the UK, such as in Australasia and Canada.[26] While the survey was distributed across each UK nation, this was either through the College of Paramedics or on social media—and thus paramedics not registered with the professional body, or not on social media, may not have had access. At best, the surveyed respondents constitute one-third of the paramedic workforce in primary are, and therefore, results should not be generalised to the entire population of paramedics working in the primary care.

It is noted that the uptake of the survey in Northern Ireland and Scotland was low. The number of paramedics working in primary care roles is likely to be fewer than

in England, however, the paramedic role is currently not captured in workforce data for these countries,[27 28] so the actual number of paramedics in primary care in these countries is unknown. However, there was no variation by region reported in the clinical work undertaken, outlining that paramedics are working in a similar way across the UK. Nevertheless, we recognise that paramedics from Northern Ireland and Scotland may be underrepresented in this survey.

The use of data triangulation within the cross-sectional survey allowed for the exploration of both the distribution and variety of roles paramedics undertake in primary care. Additional free-text responses also enabled further understanding of the support and frustrations paramedics experience when working in primary care. However, in the interpretation of the quantitative data we undertook multiple statistical tests. While adjustments were made to control familywise error rate, it is possible that some of the marginal results produced are not robust. Further research, using different variable types, could build on these findings to determine their strength.

It is appreciated that data collection took part during the ongoing global pandemic of COVID-19. As this is the first national survey of the paramedic role in primary care, it is unclear what impact this ongoing pandemic has on these results. Future surveys may be able to determine this.

## RECOMMENDATIONS

Based on our findings above, we make a number of recommendations:

### Education and support

► There is a need to further standardise required education and training for paramedics working in primary care.
► There should be systems in place within primary care settings to ensure paramedics have access to structured clinical supervision.
► Support should be given to employers to enable them to effectively support the paramedic in primary care and use this healthcare professional to the greatest benefit.

### Governance

► There is a need to standardise job titles, salaries and descriptions for paramedics working in primary care to ensure role recognition by both patients and the wider primary care team.
► A standard scope of role for paramedics working in primary care that is applicable to all nations is needed.
► Clear career pathways for paramedics working in primary care should be established, to maximise retention and job satisfaction in this clinical setting.

### Changes to legislation

► Changes to legislation to support independent prescribing of controlled medications.
► Ability for paramedics to provide a Statement of Fitness for Work.

## CONCLUSION

As the first survey of the paramedic role in primary care across the UK, this study indicates the demographic range of paramedics working in primary care, and the common clinical presentations and examinations undertaken by this workforce. As well as offering an insight into the clinical scope of role undertaken in this setting, we have highlighted that relationships exist between paramedic clinical exposure in primary care, level of education and ability of independently prescribe medicines and the extent to which clinical presentations are seen and examinations performed. As well as policy makers, this is important information for primary care employers seeking to employ, or develop, paramedics in this practice setting.

**Acknowledgements** We would like to thank the patient and public group involved with this research for their time and comments during the design of the survey and the analysis of the results. We would also like to thank our stakeholders from the College of Paramedics, Health Education England, the Nuffield Trust and the Royal College of General Practitioners with whom we discussed the development of our survey, and with whom we piloted the survey. Lastly, thank you to the paramedics across the UK who completed this survey, and the peer reviewers of this paper prior to publication.

**Contributors** GE, ST, GW, KRM and VW designed the survey. Data analysis was carried out by GE, JO and ST. The manuscript was drafted by GE, ST, GW, JO, VW and KRM. All authors accept full responsibility for the work and/or the conduct of the study, had access to the data and controlled the decision to publish. All authors read and approved the final manuscript.

**Funding** GE is supported by a National Institute for Health Research (NIHR) Doctoral Research Fellowship (NIHR300681). This work was also supported by NHS Health Education England (ref: 190121).

**Disclaimer** The views expressed are those of the authors and not necessarily those of the NHS, the NIHR, NHS Health Education England or the host institution. The funders of this research remained independent of the project design, data collection and data analysis. GE accepts full responsibility for the finished work and/or the conduct of the study, had access to the data, and controlled the decision to publish.

**Competing interests** GE is a Trustee in the College of Paramedics. The recruitment of paramedics to the survey was made by application to the College of Paramedics Research Design Advisory Committee and was not associated with GE's trusteeship. The remaining authors declare no competing interests.

**Patient and public involvement** Patients and/or the public were involved in the design, or conduct, or reporting, or dissemination plans of this research. Refer to the Methods section for further details.

**Patient consent for publication** Not applicable.

**Ethics approval** This study was reviewed by the University of Oxford Central University Research Ethics Committee (MS IDREC Ref: R64129/RE001). Participants gave informed consent to participate in the study before taking part.

**Provenance and peer review** Not commissioned; externally peer reviewed.

**Data availability statement** Data are available on reasonable request.

**ORCID iD**
Georgette Eaton http://orcid.org/0000-0001-9421-2845

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
