## [Reviewer comments · BMJ Open]

ARTICLE DETAILS

TITLE (PROVISIONAL)	Understanding the roles and work of paramedics in primary care: A national cross-sectional survey
AUTHORS	Eaton, Georgette; Tierney, Stephanie; Wong, Geoff; Oke, Jason; Williams, Veronika; Mahtani, Kamal

VERSION 1 – REVIEW

REVIEWER	Eaton-Williams, Peter South East Coast Ambulance Service NHS Foundation Trust, Research and Development
REVIEW RETURNED	12-Sep-2022

GENERAL COMMENTS	Thank you for asking me to review this accomplished article describing a well-conducted observational study of paramedics working in primary care. I believe that the authors have provided important insight into the scope of roles currently being performed in that setting. The mixed methods of analysis are a strength of the study and I was also particularly interested in how the study used HEE's roadmap of presentations and examinations to reveal participants' scope of practice. The study's objectives are clearly set out and the results are well presented in an accessible format. Both Discussion and Conclusion seem appropriate. I have indicated above that I do not believe the abstract to be complete simply because the use of mixed methods of data analysis is not declared explicitly within it. I have indicated above that a specialist statistical review is required because I am not qualified to review the figures given myself. How this is resolved must be an editorial decision. Beyond that, I am limited to raising points on individual sentences: P4.L22 the figure provided for Wales is lacking a percentage sign. P6.L.7 ...increasingly taken up clinical employment away from ambulance services, moving into primary care... NB. many have also gone to ED departments, perhaps 'many moving'? P6.L10. Changes to healthcare access has created... 'have' not 'has'? P9.L.13 Based on workforce data published for the period within which the survey was undertaken (September-November 2021) (16,17), these figures represent 33% of the population for England and Wales separately." I am struggling to understand this sentence when it follows a single figure only related to England. Perhaps 'our participants' rather than 'these figures'? P10.L5 "The majority (52%) of respondents were educated to at least one module at FHEQ Level 7/SCQF Level 11, with more than half educated to postgraduate certificate (n=90)." I believe that this means more than half of that 52% not the whole sample, but it is slightly confusing. Would 'half of those..' benefit clarity? P11.L29. Job title made a difference to the extent to which a
--

	diagnosis is made during a consultation ($H(3)=15.73$, $p<.001$), the management of medical and clinical complexity ($H(3)=15.73$, $p<.001$),....Looking at supplemental table A I don't believe these values are correctly identical, can you check this please? P16.L27. Experiences of patient dissatisfaction following consultation by a paramedic in primary care was also reported by respondents. This occurred... 'were' and 'these'? P20.L31 RECOMMENDATIONS Based on our findings above we make a number of recommendations: 'of' not 'or'. P23 Acknowledgements could include the participants? P26 reference 15 states LODNON. Those are the only points I have identified from reviewing this well written and informative article.
--	---

REVIEWER	Barry, Tomás University College Dublin, School of Medicine
REVIEW RETURNED	14-Sep-2022

GENERAL COMMENTS	Thank you for the opportunity to review this manuscript. The authors set out to provide a national (UK wide) picture of the paramedic role in primary care. This is an important aim as it addresses a key knowledge gap. To my mind a key limitation of the exercise is the response rate. Outside of England the number of respondents are small. Some estimation of denominators might be provided to help readers make an assessment of rough response rate per region. In my opinion, in the results section, it would be useful to focus more on a descriptive analysis of the results from sections 1-3 and the yes-no questions from section 5 of the survey tool. There are a lot of very interesting questions asked that together can provide a descriptive profile of paramedics working in primary care in the UK and their activities. Tables and graphs could provide key summaries and could describe/summarize the various relevant proportions in various categories. I would suggest that throughout the results section it would be useful to routinely include % and numerator / valid denominator for clarity. As I understand it care is needed with multiple statistical testing and the interpretation of p values in this context. In the first instance I would suggest that a tabulated comparison of relevant proportions etc might be most informative and this could be accompanied with statistical testing where this is considered appropriate (but ?? should be adjusted for multiple testing). I'm open to correction but would the sample size metric detailed on pg 8 line 5 not assume random sampling and a single outcome of interest? The frequency of various presentation and clinical actions described in Fig 4 and 5 would appear to be important results. I am not entirely clear though how these tables were arrived at. Respondents are asked to rate each item on a likert type scale from not at all to all the time. Should the proportions reporting each frequency per item not be reported? It might also be important to consider the results in the context of baseline frequency of said item in general in primary care. In terms of the correlational analysis from 'core capabilities of paramedics working in primary care' (Table A) on. My understanding is that in table A you are exploring the correlation of the proportion of respondents reporting a HEE core capability at various frequencies
---

	and a variety of other (ordinal) variables that could form some measure of seniority. I am again open to correction here, but scanning table A the only strong correlation I see is independent prescribing correlated with prescribing status. I'm not sure ultimately what this analysis adds and whether the statement made about positive relationships existing is justified by the analysis presented? I think the same might also apply to the other correlation tables. I would wonder if this is the most appropriate approach to analysis. I would suggest consider separating out and clearly labelling quantitative and qualitative results for clarity. Can the absence of a comment in a free text field be taken to support or deny a sentiment? I'm not sure it can... Some care may be needed here. For instance p10 line 45..'only 25 respondents had a clinical supervision model that was regular and structured' How was this statement arrived at? The qualitative data can however certainly add depth to the other data presented. Other Comments Pg 7, line 15, ref 14 ? inappropriate ref Consider including a brief description of paramedic education pathways/ pathways to/ prep for primary care for international readers. Consider including more concrete detail on how PPI/Stakeholders shaped survey instrument/ were involved in analysis. In summary, I wonder if a greater focus on a descriptive analysis with some additional targeted statistical testing might be a more appropriate approach to address the study aims. The research question is important and valuable data has been collected.
--	--

REVIEWER	Majchrowska, Anita Medical University of Lublin, Chair and Department of Humanities and Social Medicine
REVIEW RETURNED	15-Sep-2022

GENERAL COMMENTS	This is an interesting paper on a timely topic. The survey of UK paramedics working in primary care is useful and fairly detailed. I think the paper has potential to make a useful contribution to knowledge referring to health care system functioning. However, it also has several shortcomings, all of which have to do with the interpretation of the survey and the overall purpose of the paper; both of which could be clearer. Abstract The aim of the study specified in Abstract part, was to understand the role of paramedics in primary care. The research question seems too general. It needs to be operationalized. What does it mean 'to understand'? What are the criteria, etc? In consequence of the vagueness of the Abstract, Conclusions part (in the Abstract) is not consistent with the aim of the research. Methods The authors write about qualitative and quantitative (mixed-method) research conducted among paramedics (lines 47-52, p. 7). The description should be clearer, specifically indicating the way of different data analysis. A detailed explanation of the methodology (data collection and analysis) will help the readers to better understand research methods (whether a combination of two
---

	methods was implemented or just one specific method of data collection). It might be interpreted that the author/s have used just one method, which is survey, but tried to get help from qualitative data in addition to quantitative through combination of closed and open-ended questions to enrich the findings. This is a kind of data triangulation i.e. using more than one source of data in a single study not mixed methods design. If the participation of patients and other groups of the studies (at several stages of this research) is indicated, this participation should be described in more details (p. 8, lines 3-7). As the strength of this study, the authors point out that this is the first national survey conducted among paramedics working in primary care within UK. However, we should remember that the sample was not representative and they have used purpose-sampling (as indicated by the "snowball" method). Those results must not be generalized in any way to the entire population of paramedics working in the 'primary care'. As the authors wrote, the surveyed respondents constituted about 1/3 of the paramedic employed in that area. There is a wide range of data presented in this manuscript and described by the authors. For this reason Discussion section should be broadened, with more items of references, regarding discussed issues. Conclusions I am not sure if all the statements in Conclusion part are consistent with the result (e.g. regarding patients group). Overall, I enjoyed reading the paper and I think it is worth recommending after the revision. I wish the authors the best of luck with the paper.
--	--

VERSION 1 – AUTHOR RESPONSE

Thank you, Reviewer 2, for your comments.

We have considered your comment:

To my mind a key limitation of the exercise is the response rate. Outside of England the number of respondents are small. Some estimation of denominators might be provided to help readers make an assessment of rough response rate per region.

We appreciate the limitation that the response rate for Northern Ireland and Scotland places on this research, however without workforce figures for paramedics in primary care in these countries, we cannot provide an accurate denominator to enable comparison. We are concerned that any estimations would be misleading to readers.

Thank you for your comment:

In my opinion, in the results section, it would be useful to focus more on a descriptive analysis of the results from sections 1-3 and the yes-no questions from section 5 of the survey tool. There are a lot of very interesting questions asked that together can provide a descriptive profile of paramedics working in primary care in the UK and their activities. Tables and graphs could provide key summaries and could describe/summarize the various relevant proportions in various categories. I would suggest that throughout the results section it would be useful to routinely include % and numerator / valid denominator for clarity.

We have considered this in our revision of the text. We are, however, restricted in our word count for reporting the volume of results in this survey – and have therefore chosen to focus on those results most pertinent to present an overview paper.

Thank you for your comment:

As I understand it care is needed with multiple statistical testing and the interpretation of p values in this context. In the first instance I would suggest that a tabulated comparison of relevant proportions etc might be most informative and this could be accompanied with statistical testing where this is considered appropriate (but ?? should be adjusted for multiple testing). I'm open to correction but would the sample size metric detailed on pg 8 line 5 not assume random sampling and a single outcome of interest?

Thank you for highlighting the need to clarify our statistical results. We did adjust for multiple comparisons in our results, and we have made this clearer in the main manuscript, as well as in the reporting of the results within Tables A, B, C and D in online supplemental file 2.

We have considered your comment:

The frequency of various presentation and clinical actions described in Fig 4 and 5 would appear to be important results. I am not entirely clear though how these tables were arrived at. Respondents are asked to rate each item on a likert type scale from not at all to all the time. Should the proportions reporting each frequency per item not be reported? It might also be important to consider the results in the context of baseline frequency of said item in general in primary care.

Figure 4 and 5 are treemaps, used to display hierarchical data in nested figures. We considered this an appropriate way to display the frequencies resulting from the questions that used a Likert type scale. We opted for treemaps over other types of graph as treemaps can be used to legibly display several items on the screen simultaneously, which is an efficient use of space in an electronic journal article. We have amended the title of both figures to confirm that they are treemaps, and therefore confirm to readers a tiling algorithm was used for their production.

Thank you for your suggestion:

Can the absence of a comment in a free text field be taken to support or deny a sentiment? I'm not sure it can... Some care may be needed here. For instance p10 line 45.. 'only 25 respondents had a clinical supervision model that was regular and structured' How was this statement arrived at? The qualitative data can however certainly add depth to the other data presented.

On P10, L45 we reported the respondents who explicitly reported a type of clinical supervision model. However, we acknowledge the bias that the sentiment 'only' adds to this sentence. We have revised this paragraph to avoid such an assumption.

We have addressed your minor comments concerning P7, L15 (reference 14) and have amended the reference list as necessary. Thank you for spotting our oversight in regard to this.

We have considered your suggestion:

Consider including a brief description of paramedic education pathways/ pathways to/ prep for primary care for international readers.

However, due to a limited word count for this article, we feel it is out of the scope of this paper to give this overview. Appropriate references are used where paramedic education and frameworks for work in primary care are included in the text, and it is our hope that international readers would consider these in their reading of this paper.

Thank you for your comment:

Consider including more concrete detail on how PPI/Stakeholders shaped survey instrument/ were involved in analysis.

We have added more detail to this component of the paper, whilst still considering the overview word count of the article.

Finally, we have reflected on your comment:

In summary, I wonder if a greater focus on a descriptive analysis with some additional targeted statistical testing might be a more appropriate approach to address the study aims. The research question is important and valuable data has been collected.

We believe we have already presented the paper in this way. However, we have made more explicit the statistical testing that has been undertaken, and the relevance of this in the claims we make. We hope these revisions address your comment.

Thank you, Reviewer 3, for your comments.

The aim of the study specified in Abstract part, was to understand the role of paramedics in primary care. The research question seems too general. It needs to be operationalized. What does it mean 'to understand'? What are the criteria, etc? In consequence of the vagueness of the Abstract, Conclusions part (in the Abstract) is not consistent with the aim of the research.

Thank you for highlighting our shortcomings regarding this important component of the paper. We have amended the abstract to relate specifically to the objectives listed in the research paper and avoided the use of general terminology in this revision.

Thank you for your suggestion:

The description should be clearer, specifically indicating the way of different data analysis. A detailed explanation of the methodology (data collection and analysis) will help the readers to better understand research methods (whether a combination of two methods was implemented or just one specific method of data collection).

We have revised our methodology section to make clearer the data triangulation approach we used in this survey.

Thank you for your suggestion:

If the participation of patients and other groups of the studies (at several stages of this research) is indicated, this participation should be described in more details (p. 8, lines 3-7).

We have added more detail to this component of the paper, whilst still considering the overview word count of the article.

Thank you for your comment:

As the strength of this study, the authors point out that this is the first national survey conducted among paramedics working in primary care within UK. However, we should remember that the sample was not representative and they have used purpose-sampling (as indicated by the "snowball" method). Those results must not be generalized in any way to the entire population of paramedics working in the 'primary care'. As the authors wrote, the surveyed respondents constituted about 1/3 of the paramedic employed in that area.

We have made this explicit in the limitations section of our survey.

We have considered your suggestion:

There is a wide range of data presented in this manuscript and described by the authors. For this reason Discussion section should be broadened, with more items of references, regarding discussed issues.

We consider that we have presented an overview paper of the research undertaken. In doing so, we have not been able to add further breadth to the discussion section, whilst still remaining in the word count limits for research published within BMJ Open.

Lastly, thank you for your comment:

I am not sure if all the statements in Conclusion part are consistent with the result (e.g. regarding patients group).

We have amended this section for clarity and to ensure that it remains consistent with the results presented in the paper.